# Is There a Difference in the Utilisation of Inpatient Services Between Two Typical Payment Methods of Health Insurance? Evidence from the New Rural Cooperative Medical Scheme in China

**DOI:** 10.3390/ijerph16081410

**Published:** 2019-04-19

**Authors:** Dai Su, Yingchun Chen, Hongxia Gao, Haomiao Li, Jingjing Chang, Shihan Lei, Di Jiang, Xiaomei Hu, Min Tan, Zhifang Chen

**Affiliations:** 1School of Medicine and Health Management, Tongji Medical College, Huazhong University of Science and Technology, Wuhan 430030, China; sudai@hust.edu.cn (D.S.); gaohongxia@hust.edu.cn (H.G.); lihaomiao@hust.edu.cn (H.L.); changjingjing@hust.edu.cn (J.C.); leishihan@hust.edu.cn (S.L.); jiangdi@hust.edu.cn (D.J.); huxiaomei@hust.edu.cn (X.H.); tanmin@hust.edu.cn (M.T.); chenzhifang@hust.edu.cn (Z.C.); 2Research Center for Rural Health Services, Hubei Province Key Research Institute of Humanities and Social Sciences, Wuhan 430030, China

**Keywords:** difference, utilisation of inpatient services, payment method, new rural cooperative medical scheme, interrupted time-series analysis, propensity score matching, China

## Abstract

This study aimed to evaluate the effects of the differences between two typical payment methods for the new rural cooperative medical scheme (NRCMS) in China on the utilisation of inpatient services. Interrupted time-series analysis (ITSA) and propensity score matching (PSM) were used to measure the difference between two typical payment methods for the NRCMS with regard to the utilisation of inpatient services. After the reform was formally implemented, the level and slope difference after reform compared with pre-intervention (distribution of inpatients in county hospitals (DIC), distribution of inpatients in township hospitals (DIT) and the actual compensation ratio of inpatients (ARCI)) were not statistically significant. Kernel matching obtained better results in reducing the mean and median of the absolute standardised bias of covariates of appropriateness of admission (AA), appropriateness of disease (AD). The difference in AA and AD of the matched inpatients between two groups was −0.03 (*p*-value = 0.042, 95% CI: −0.08 to 0.02) and 0.21 (*p*-value < 0.001, 95% CI: −0.17 to 0.25), respectively. The differences in the utilisation of inpatient services may arise owing to the system designs of different payment methods for NRCMS in China. The causes of these differences can be used to guide inpatients to better use medical services, through the transformation and integration of payment systems.

## 1. Introduction

The medical service system in rural areas in China is usually defined as the region where the three-level rural medical service network is located, namely the county scope, which includes the three-level rural medical institutions in the county, and the service objects should be consumers within the county scope. Together with economic development, health insurance has been increasingly developed worldwide. Undoubtedly, the payment method for health insurance has an extremely important role in health demand and fund protection, especially for rural areas. Therefore, utilisation of inpatient services is a key factor in assessing the effectivity of the payment method for health insurance [1]. Evidence from many countries, such as South Africa, South Korea and Ghana [2,3,4], has shown that the utilisation of inpatient services is influenced by not only supply constraints but also demand constraints. Such utilisation was well adjusted by optimising the behaviour of the supply and demand sides based on the reform of a single health insurance payment method. However, the difference between different payment methods on the utilisation of inpatient services remains unclear [5,6]. Hence, this research particularly focuses on the difference in the utilisation of inpatient services between two typical payment methods for NRCMS in China, which plays a guiding role in the improvement of payment methods.

The increased funding and reimbursement for the NRCMS provided rural population with improved access to inpatient services and remarkably promoted the utilisation of inpatient services. The annual hospitalisation rate of rural residents for NRCMS rose from 3.4% in 2003 to 9.0% in 2013, then reached 17.6% in 2017 [7]. However, the problem of the utilisation of inpatient services in rural China remains prominent, especially hospital choices and health expenses. Regardless of the severity of disease, rural inpatients may choose high-level hospitals with relatively abundant medical resources, which may weaken the functions of hospitals at various levels. In 2013, the proportion of inpatients for NRCMS in county hospitals are 54.4%, whereas that in township hospitals is only 29.5% [8,9]. Meanwhile, this situation may obviously increase the out-of-pocket (OOP) and total expenses of inpatients. The effective reimbursement ratio of inpatient services was lower than the nominal reimbursement ratio that was originally designed by the NRCMS and will thus reduce the efficiency of fund for NRCMS.

To solve the problems, the Chinese government promoted the reform of the payment method for NRCMS nationwide, which was piloted in 2004 and fully implemented in 2012. This reform experienced a transition from single to mixed payment method, especially in the case of the global budget in conjunction with other payment methods, which is the main direction of the reform of payment system for NRCMS in China. In rural China, there are mainly two types of innovative payment methods for the global budget for NRCMS, namely, hospital-based payment and regional or medical alliance-based payment. Each payment method has changed from fee-for-diseases to diagnosis related groups (DRGs). However, these two payment methods differ substantially in fund management and allocation and per unit payment. Specifically, regional or medical alliance-based payment means that the fund management department prepays the fund into a county-level hospital with the maximum medical service capability in the region or medical alliance. In addition, it is responsible for the benefit allocation among hospitals in the alliance. It provides a fund management system. The hospital-based payment method represents a mixed payment method for a single hospital in the county. The management and allocation of the fund remain confined within a single hospital. In addition, different rural areas in China developed specific details in the two typical payment method reforms. The reform of payment methods for NRCMS in China is mainly confined to a single hospital, and its role is more reflected on the control of medical expenses in a single hospital. Thus, the improvement of the utilisation of inpatient services for rural residents is limited.

Recently, several studies in China have explored the effect of the two typical payment type reforms for NRCMS on the utilisation of inpatient services. Qian [10] selected the payment reform in Ningxia, which is hospital-based and fee-for-disease, as a sample and found that inpatients prefer to select county hospitals. Thus, patients with diseases that can be cured in township hospitals opt to go to county-level hospitals, where the actual medical cost is not alleviated. Li [11] assessed quota payment for specific diseases under global budget mode which is one of the most typical modes of payment system reform in rural China from aspects of the total fee, structure of the fee and enrollees’ benefits, and found that specific diseases under global budget had obviously positive effects on cost control in Weiyuan, Gansu. He [12] chose YC county as the intervention group and ZJ county acted as the control group, found that the hospital-based global budget has a prominent impact on curbing the growth of insurance fund expenditures, as well as drug and medical consumable costs. However, the patients’ out-of-pocket payment has risen. Si [13] selected Anhui, with medical alliance-based payment and fee-for-diseases, and indicated that the payment reform enhanced the ability of primary care and promoted inpatients to township hospitals. However, the reform may have also promoted the risk of inappropriate admission, and doctors may induce inpatient demand based on information asymmetry. 

Previous studies indicated that different payment methods have specific effects on the utilisation of inpatient services. However, these researches are only limited to the policy effect of single payment methods. Thus, the lack of comparison of different payment methods provides a basis for optimising the utilisation of inpatient services. Meanwhile, some methods in related research mainly focus on descriptive statistics and lack rigorous measurement statistics [14,15,16].

Therefore, this study proposes to explore the disparities of the utilisation of inpatient services amongst two payment reform groups for NRCMS in China. Two relatively rigorous and innovative methods in the field of health services were used to examine the above differences, namely, ITSA and PSM. Hence, we hypothesised that the difference between two payment methods for NRCMS on inpatient service utilisation exists owing to the specific policy design. Then, we provide suggestions on the practicability of the two payment reform methods for NRCMS and means to promote and ameliorate it.

## 2. Materials and Methods

### 2.1. Data Source and Sampling

We aimed to ensure that the sampling areas can represent the main forms and characteristics of payment systems for NRCMS in China. Therefore, two counties were selected as samples (Dingyuan in Anhui Province and Weiyuan in Gansu Province) according to the two forms of global budget in eastern and western China, respectively. Table 1 shows the basic information about the rural residents’ economic status and health development in two counties. The starting time for the reform of the payment method for NRCMS in these two counties is the same, which is an important factor in selecting these two counties. Meanwhile, differences in the design of the specific payment methods between the two counties are significant (Table 2).

Data were derived from two databases, one is the National Dataset for NRCMS in Dingyuan and Weiyuan. The payment reform in two counties was formally launched in 2016. 343,981 and 187,603 hospitalisation records between 1 January 2014 and 30 June 2017 for NRCMS in Dingyuan and Weiyuan, respectively, were extracted from the database. These data were quality-checked by the National Health and Family Planning Commission and hospitals of Dingyuan and Weiyuan. Another database consists of electronic health records (EHRs) in hospitals. All inpatient information, such as address and inpatient number, was excluded before the data were provided to the study team. In each county, one county-level hospital and three township-level hospitals were selected as our sample hospitals.

At the county level, we selected the largest hospital in the county as our sample hospital. At the township level, we selected three township-level hospitals according to distance from the county and population density to ensure that the sample hospitals had sufficient sample sizes and maximise the effect of the reform of the payment system for NRCMS. Cluster sampling was used to select 300 EHRs in each hospital, which were taken every year from 2015 to 2017. In total, 3245 EHRs were admitted from the database and then the remaining 3036 EHRs after selection (Figure 1).

### 2.2. Outcome Variables

This study screened the outcome variables of the previous research on the utilisation of inpatient services in the payment method and abandoned the traditional outcome variables of the use of inpatient services, such as hospitalisation rate, that is, the rate of need-but-not inpatient care. According to the characteristics of payment reform, the following five outcome variables were selected to reflect the utilisation of inpatient services, namely, distribution of inpatients in county and township hospitals (DIC and DIT, respectively), actual compensation ratio of inpatients (ACRI), appropriateness of admission (AA) and appropriateness of disease (AD). Descriptions of these outcome variables are as detailed as follows.

#### 2.2.1. Behaviour of Inpatient Service Utilisation

Three outcome variables, DIC, DIT and ACRI, were used to represent the impact of payment system reform on inpatient service utilisation behaviour. These outcome variables were derived from the National Dataset for NRCMS in Dingyuan and Weiyuan and calculated as follows: (1) DIC/DIT (constituent ratio of inpatients in CHs and THs): number of inpatients in CHs/THs divided by the total number of inpatients and (2) ACRI: inpatient compensation fee of NRCMS funds divided by the total hospitalisation expenses.

#### 2.2.2. Appropriateness of Inpatient Service Utilisation

The AA and AD indicators were derived from admission records and identified by professional assessment tools. The appropriateness evaluation protocol (AEP) is widely used in the evaluation of AA. Zhang and Chen [17] developed an identification tool, namely, R-AEP, which is suitable for the AA to hospitals that are operating in rural areas in China. The tool is based on the existing international AEP and provides a methodological basis for evaluation. The R-AEP criteria were used in the evaluation of the AA in admission records. Evaluation was performed by professionally trained personnel. The R-AEP criteria are based on the value of the admission record. Therefore, all R-AEP-related indicators were extracted and their actual values were compared with standard values. The actual values of the indicators that match the standard values were considered appropriate. Admission records with all relevant values that did not meet the AEP criteria were considered inappropriate admissions.

Dingyuan and Weiyuan implemented pay-for-diseases at the county and township hospitals and prescribed a list of diseases that should be treated according to hospital service capabilities. The judgment of AD is based on this list of diseases. If the disease is included in the corresponding hospital list, then it is considered reasonable. Otherwise, it is unreasonable.

### 2.3. Statistical Analysis

To estimate the difference of effects of payment reform on the utilisation of inpatient services in China, we set Dingyuan as the treatment group and Weiyuan as the control group and compared the outcomes between the two groups. ITSA was used for three outcome variables, namely, DIC, DIT and ACRI, which were extracted from hospitalisation records. The two other outcome variables, namely, AA and AD, which were derived from the admission case database after sampling, were analysed by using the PSM method to increase the reliability of the results.

#### 2.3.1. Interrupted Time Series Analysis

ITSA is divided into single- and multiple-group analyses. In this paper, single-group ITSA compares the level and trend before and after intervention to evaluate the effect of single intervention [18,19]. Multiple-group ITSA was used to effectively evaluate the long-term effects of intervention between the two groups and may be particularly valuable when exogenous policy shift exists, which may affect all groups. It comprehensively considers the original trends of variables and compares differences in the level and trend before and after intervention and between the two groups [20]. ITSA is proposed as a flexible and rapid design to be considered before defaulting to traditional two-arm, randomly controlled trials. [21,22]

The standard regression models of the single—(1) and multiple-group (2) ITSA are as follows:(1)$$Y_{t}=\beta_{0}+\beta_{1}T_{t}+\beta_{2}X_{t}+\beta_{3}X_{t}T_{t}+\varepsilon_{t}$$
(2)$$Y_{t}=\beta_{0}+\beta_{1}T_{t}+\beta_{2}X_{t}+\beta_{3}X_{t}T_{t}+\beta_{4}Z+\beta_{5}ZT_{t}+\beta_{6}ZX_{t}+\beta_{7}ZX_{t}T_{t}+\varepsilon_{t}$$
where ***Y_t_*** is the mean number of outcome variables measured and an evaluation index that describes the study objects in month ***t***, ***T_t_*** is a time series variable that indicates the time in months at time t from the start of the observation period (coded as 1, 2, 3, 4… up to the last month), ***X_t_*** is a dummy variable that indicates time ***t***, which is assigned a value of ‘0’ if time t occurred before policy intervention and ‘1’ if time t occurred after policy intervention, and ***X_t_T_t_*** is an interaction term that is calculated as (*T* − 19) × *X*, such that it runs sequentially starting at 1. Furthermore, ***Z*** is a dummy variable that indicates treatment status (***Z*** = 1 and ***Z*** = 0 for the treatment and control groups, respectively). $\varepsilon_{t}$ is the random error term at time t, which represents an unknown component of the model. In the single-group model, *β*_0_ estimates the intercept or baseline level of the outcome variable per month at the start time of the observation period, *β*_1_ estimates the slope or trend of the outcome variable before policy intervention, *β*_2_ estimates the level change of the outcome variable immediately following the introduction of policy intervention and *β*_3_ estimates the slope or trend change of the outcome variable between pre- and post-intervention. However, in the multiple-group model *β*_0_ to *β*_3_ represent the control group, and *β*_4_ to *β*_7_ represent the values of the treatment group. Specifically, *β*_4_ represents the difference in the level (intercept) of the outcome variable between the treatment and control groups pre-intervention; *β*_5_ represents the difference in the slope (trend) of the outcome variable between the treatment and control groups prior to the intervention; *β*_6_ indicates the difference between the treatment and control groups in the level of the outcome variable immediately following the introduction of the intervention and *β*_7_ represents the difference between the treatment and control groups in the slope (trend) of the outcome variable after the initiation of the intervention compared with pre-intervention (akin to a difference-in-differences of slopes).

We used a monthly mean value of outcome variables and assigned January, 2016 as the intervention time point for the formal implementation of the payment reform in the two counties. Segmented linear regression divides the time series into pre- and post-January 2016 segments. Seasonality adjustment was not required because the autocorrelation function of the outcome variables from 1 July 2014 to 30 June 2017 was tested. To allow for autocorrelation in the data, the official Stata packages ‘newey’ and ‘prais’ were used to fit the generalised least-squares (GLS) regression. Two packages allow for fitting a segmented linear regression model under the condition of autocorrelation and controlling for confounding omitted variables. We obtained the order of autocorrelation by examining the autocorrelation and partial autocorrelation functions [21]. The Durbin–Watson (DW) statistic was used to test whether the random error terms follow a first-order autoregressive [AR (1)] process [22,23]. The DW value was 1.4385, and autocorrelated disturbances existed.

#### 2.3.2. Propensity Score Matching

Several characteristics of the control group may be inconsistent with those of the treatment group. To address this shortcoming, we constructed an appropriate control group by using PSM, which is widely used in estimating the effects of health and other policy interventions, whereas randomized controlled trials (RCTs) are not feasible. PSM can identify individuals in the control group with similar characteristics as those affected by the policy. The following covariates are used: gender, age, occupation, marital status, medical department, admission route, length of stay, admission status, health status, history of disease, chronic, history of surgery and hospital level.

Before estimating the propensity scores based on a rich set of covariates from logistic regression, we initially restricted the potential control group by considering the key individual characteristics of the treatment group and availability of data prior policy implementation. We aimed to test the balancing property of each observed covariate and overall balance between the treatment and control groups and verify the reduced sampling bias achieved through matching. The following matching methods were used: kernel matching, k-nearest neighbour matching (k = 1) and local linear regression matching. We used a with replacement kernel matching at a bandwidth of 0.06 with an epan-type kernel. The local linear regression matching required tricubic kernel with a bandwidth of 0.8. These kernel types and bandwidths are frequently mentioned in the literature [24,25,26]. Observations that deviated from the common support were excluded from the analysis. Finally, we checked the mean and median reduction biases of overall balance [27].

Assume that each inpatient *i* has two potential outcomes after matching, namely, *Y_i_*_1_ (treatment, Dingyuan) and *Y_i_*_0_ (control, Weiyuan). The average effect of the treatment is given by *E*(*Y_i_*_1_ − *Y_i_*_0_). However, *Y_i_*_0_ and *Y_i_*_1_ cannot be observed simultaneously for the inpatient in the treatment group. Instead, the average treatment effect on the treated (ATT) is calculated after matching as follows:(3)$$ATT=E(Y_{i1}|D_{i}=1)-E(Y_{i0}|D_{i}=0)$$where *D_i_* is the dichotomous indicator of treatment. A value of 1 indicates that inpatient i received the payment reform in Dingyuan and 0 in Weiyuan. Stata 14.0 software (Stata Corp LP, College Station, TX, USA) was used for statistical analysis in a Windows environment. The two-sided statistical significance level was set at 0.05.

## 3. Results

### 3.1. ITSA on the Behaviour of Inpatient Service Utilisation Before and After Reform

We firstly compare the changes in the level and trend of behaviour of inpatient service utilisation before and after the reform in each group. Figure 2 and Table 3 show that the DIC in two counties showed a slightly increasing trend after the reform, but the trend change is not statistically significant. 

After the formal implementation of the reform, DIT in Dingyuan County changed significantly and showed an increasing trend (0.24, 95% CI: 0.04 to 0.45, *p*-value < 0.05). Similarly, ARCI in Dingyuan County increased significantly (0.89, 95% CI: 0.60 to 1.19, *p*-value < 0.001). However, the DIT and ARCI in Weiyuan County remained unchanged.

### 3.2. ITSA on the Differences in Behaviour of Inpatient Service Utilisation Between Groups

The key assumption of multiple-group ITSA requires a change in the level or trend in the outcome variable, which is presumed to be the same in the two groups. Therefore, as shown in Figure 2 and Table 3, we also examined and found that the differences in the level and slope prior to the reform were not statistically significant. Nevertheless, differences between two groups in the period immediately following reform in level and slope compared with pre-intervention were not statistically significant.

### 3.3. Characteristics of Our Study Sample for PSM Method

Table 4 presents the basic characteristics of sample inpatients of two counties after the formal implementation of the reform before matching.

### 3.4. Balance Test and Matching Results from Three Matching Methods

Table 5 reveals that the overall balancing properties can be verified by comparing the joint significance of all matching variables in the logit models before and after matching. In columns (2) and (3), the results further indicate that the three matching methods improved overall balance after matching (*p*-value > 0.005). Columns (4) and (5) consistently show that the mean and median of the absolute standardised bias were reduced substantially by the three matching methods. However, kernel matching obtained better results in reducing the mean and median of the absolute standardised bias of covariates of AA (mean bias = 2.8; median bias = 2.0), AD (mean bias = 2.2; median bias = 1.4).

As summarised in Table 5, the difference of the AA of the matched inpatients between the two groups was −0.03 by kernel matching (*p*-value = 0.042, 95% CI: −0.08 to 0.02) and −0.04 by k-nearest neighbour matching (*p*-value = 0.038, 95% CI: −0.11 to 0.03). Meanwhile, the difference of the AD of the matched inpatients between the two groups was 0.19 by kernel matching (*p*-value < 0.001, 95% CI: 0.17 to 0.25), 0.17 by k-nearest neighbour matching (*p*-value < 0.001, 95% CI: 0.15 to 0.19) and 0.16 by local linear regression matching (*p*-value < 0.001, 95% CI: 0.14 to 0.20). Such results indicate that the AA in Dingyuan is better than that in Weiyuan after matching, whereas the AD in Weiyuan is better than that in Dingyuan.

## 4. Discussion

After the reform of payment methods in two counties, DIC, DIT and ACRI showed an upward trend, which indicated that the reform of two counties has a positive effect on the behavior of inpatient services utilisation. Specifically, inpatients returned to the county from high-level hospitals outside the county, and the choice of inpatients for county and township hospitals also became increasingly appropriate, which are consistent with the research of Li and Yu [28,29] However, compared with the pre-intervention, DIT and ACRI in Weiyuan achieved significant results (increased by 0.11% and 0.54% per month). The findings of Wachter and Brown [30,31] showed that rural residents may prefer price to quality, and they more or less consider compensation ratio during decision making. The NRCMS fund pays for specific diseases under the global budget according to the disease list, although the two counties separately specified the list of appropriate treatment for a range of diseases at the county- and township-level hospitals. Although diseases may be duplicated in the two lists in Dingyuan, they are not repeated in the two lists in Weiyuan. Inpatients who should be treated in township-level hospitals will not be reimbursed (20% in 2016 and 0% in 2017) if they insist on referral to county-level hospitals, which is lower than the majority of other provinces in China. This measure fully utilises the sensitivity of inpatients to disease cost and economic burden. The compensation ratio with large differences between county and township hospitals can effectively prevent several inpatients from inappropriately selecting hospitals. The setting of differential compensation level for NRCMS uses economic leverage to limit inpatients’ tendency to bypass low-level hospitals in favour of high-level hospitals. As such, inpatients have to discover their first diagnosis at the township-level hospitals, which improved DIT in Weiyuan and improved the efficiency of the utilisation of medical resources in township-level hospitals. However, the design of the payment system in Dingyuan largely differs from that in Weiyuan. In contrast to payment for a single hospital in Weiyuan and the obvious compensation ratio for county and township hospitals, Dingyuan implements relatively loose regulations on the disease list and compensation level. However, the reform of the payment method in Weiyuan County is a mandatory intervention of the policy [24,32].

A number of scholars found that the rate of inappropriate admission in rural China in 2014 reached 15.2%, which is higher than the 9.6% reported by The Netherlands [33], 11.0% in Portsmouth [34] and 9.1% in the UK [35]. This finding indicates that the problem of AA in China is more serious compared with other countries worldwide. In this study, the AA in Dingyuan was slightly higher than that of Weiyuan (ATT = 0.03), which reflects that the reform in Dingyuan is more significant for benign changes, such as treatment behavior of the doctors and the needs of inpatients. Dingyuan prepays the NRCMS fund to the medical consortium according to capitation and packages the fund for outpatients and inpatients. The system of the referral to lead hospitals in medical consortium is an option for management and a sharing mechanism for internal benefit. The fund balance of th county- and township-level hospitals are allocated in proportions of 60%–70% and 20%–30%, respectively, which greatly promotes the rational distribution of several resources (doctors and patients) in the medical consortium. Doctors will take the initiative for outpatient treatment or referral for patients who do not meet the admission requirements to control the cost of inpatient services and reduce the reverse selection and moral hazards [36]. Such a mechanism inhibited the excessive demand and utilisation of inpatient services and maximised the benefits of hospitals, doctors and inpatients within the medical consortium. However, this mechanism is difficult to be achieved in Weiyuan, because township-level hospitals in Weiyuan are largely concerned about maximising their interests. As a result, over-hospitalisation and lack of effective supervision of NRCMS funds occur [37]. Although it is effective in optimising the hospitals’ choice of inpatient service utilisation, joint action between county- and township-level hospitals is lacking [38]. This gap is a current shortcoming of the quota standard of the global budget in Weiyuan.

In the next step of the payment system design, especially the design of the medical service system, we should focus on the improvement of healthy integration and development [39,40]. Firstly, rational referral and benign flow of medical resources are important directions for healthy integration and development in the future. In addition, the payment reform of capitation prepayment in the medical consortium is clearly advantageous. The Kaiser Permanente model of medical care in the United States can be used as a reference [41]. In the future, health services, such as medical services and prevention and rehabilitation, will be integrated to promote integration between hospitals and improve the continuity of health services. Adjusting the utilisation of inpatient services by using large differences in compensation level can effectively guide the medical supply and demand in a short period of time. Secondly, the design of the specific payment system should also be based on local specific conditions. The economic and medical developments in different regions of China largely differ. Thirdly, pay-for-diseases should developed for using DRGs, and mixed payment methods can improve the treatment process and its accuracy for NRCMS fund [42,43,44,45].

## 5. Limitations

This study has several limitations. Firstly, ITSA and PSM serve as two rigorous methods of policy effect, which may lead to biased results in this study because of inconsistencies in sample selection. Secondly, the samples were derived from the medical database using PSM. Most of the selected covariates are individual-level variables. However, other confounding factors continue to affect the outcome, which cannot be optimised because of the limitations in the database. Thirdly, in the PSM process, three matching methods were selected to evaluate the policy effect. Although these methods have good applicability in practice, other suitable matching methods may be more suitable. Finally, only the ATT of AA and AD after policy intervention is considered. The baseline data cannot be analysed because of the insufficient data, which may affect the accuracy of policy evaluation.

## 6. Conclusions

Many models for the reform of payment method for NRCMS in China exist. These payment methods have their own characteristics, and their impact on the utilisation of inpatient medical services differs significantly. Policy makers should mainly consider the differences in the effects between the current pilot areas and determine the factors that underlie these effects, which will promote the payment method for NRCMS that meets the development characteristics of the regional health care system.

## Figures and Tables

**Figure 1 ijerph-16-01410-f001:**
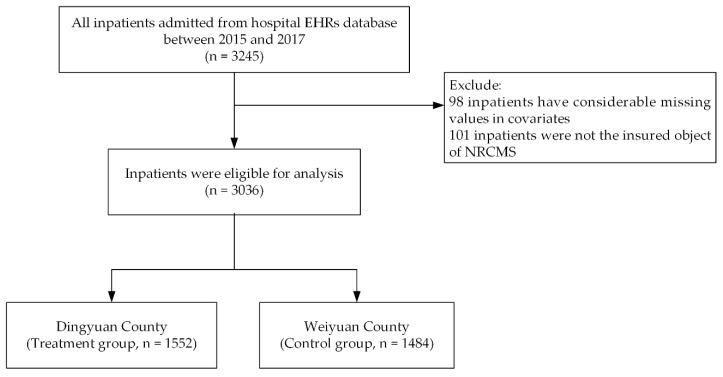
Study design and flow chart of the inpatients selection for new rural cooperative medical scheme (NRCMS) and the classify of those inpatients from hospital electronic health records (EHRs) database in two counties for propensity score matching (PSM).

**Figure 2 ijerph-16-01410-f002:**
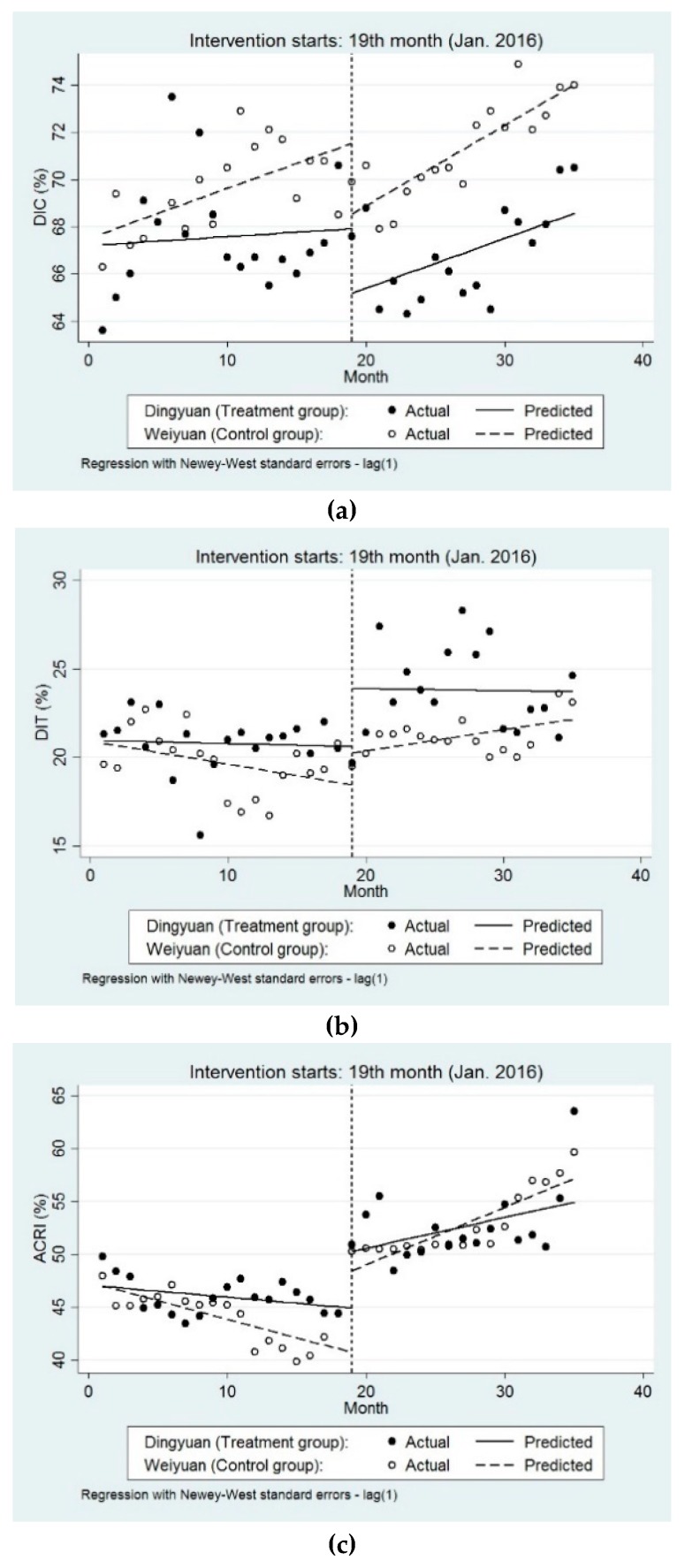
The behavior of inpatients services utilisation over time between two groups. Note: (**a**) DIC in Dingyuan and Weiyuan (%); (**b**) DIT in Dingyuan and Weiyuan (%); (**c**) ACRI in Dingyuan and Weiyuan (%).

**Table 1 ijerph-16-01410-t001:** Economic and health development information in two counties.

Characteristic	Huining County	Dingyuan County
Population (thousands)	576	982
Area (square kilometres)	6439	2998
GDP (million)	6276	18,337
Per capita disposable income of urban residents throughout the year (RMB)	17,123	23,180
No. of county-level and township-level hospitals	37	31
No. of open beds per thousand people	3.58	2.88
No. of professional physicians per thousand people	4.27	3.37
GDP, Gross Domestic Product. RMB, ren min bi.		

**Table 2 ijerph-16-01410-t002:** The differences in the specific payment methods design between two counties.

County	Dingyuan	Weiyuan
Payment type	Global budget (capitation prepayment); Payment for single disease (quota standard)	Global budget (specific diseases); Payment for single disease (quota standard)
Appropriateness treatment of disease range	150 + *N* in county hospitals, 50 + *N* in township hospitals	170 in county hospitals, 60 in central township hospitals 50 in general township hospitals
quota standard	Average cost in the past three years	Average cost in the past three years
Principle of cost reimbursement	85% for county hospitals (quota standard) 90% for township hospitals (quota standard)	If actual cost < quota standard, 70% for county hospitals (autual cost) 80% for township hospitals (autual cost); If actual cost > quota standard, 70% for county hospitals (quota standard) 80% for township hospitals (quota standard)
Single payment limitation	-	15,000 RMB for county hospitals 3,000 RMB for township hospitals
Annual Payment Limitation	200,000 RMB	80,000 RMB

*N* means that hospitals can expand other diseases range according to their medical capabilities. RMB, ren min bi.

**Table 3 ijerph-16-01410-t003:** Estimated changes and difference in level and trend of indicators of single- and two-groups comparison.

Variables	DIC (%)		DIT (%)		ACRI (%)	
	Dingyuan	Weiyuan	Dingyuan	Weiyuan	Dingyuan	Weiyuan
**Single group**						
Preintervention trend	0.04 (−0.23 to 0.30)	0.21 ** (0.07 to 0.36)	−0.02 (−0.29 to 0.03)	−0.13 (−0.29 to 0.03)	−0.11 (−0.30 to 0.07)	−0.35 ** (−0.53 to −0.16)
Level change	−2.83 (−5.95 to 0.30)	−2.30 (−4.77 to 0.15)	3.27 * (0.31 to 6.24)	1.81 (−0.37 to 4.00)	5.34 ** (0.31 to 6.24)	7.72 *** (5.00 to 10.43)
Trend change	0.18 (−0.19 to 0.56)	−0.02 (−0.24 to 0.21)	0.01 (−0.29 to 0.30)	0.24 * (0.04 to 0.45)	0.40 (−0.10 to 0.91)	0.89 *** (0.60 to 1.19)
**Two groups comparison**						
Level difference prior to intervention	−0.48 (−3.70 to 2.75)		0.14 (−2.01 to 2.30)		−0.61 (−2.68 to 2.56)	
Slope difference prior to intervention	−0.17 (−0.47 to 0.12)		0.11 (−0.08 to 0.30)		0.23 (−0.02 to 0.50)	
Level difference in the period immediately following intervention	0.26 (−3.64 to 4.17)		1.46 (−2.15 to 5.07)		−2.37 (−6.91 to 2.17)	
Slope difference after intervention compared with preintervention	0.04 (−0.36 to 0.45)		−0.24 (−0.59 to 0.11)		−0.49 (−1.06 to 0.09)	

Data in the table: β (95%CI), CI: Confidence interval; ***, **, * means statistically significant at the 0.1%, 1% and 5% levels respectively; DIC, the distribution of inpatients in county hospitals; DIT, the distribution of inpatients in township hospitals; ACRI, the actual compensation ratio of inpatients.

**Table 4 ijerph-16-01410-t004:** Characteristics of study sample after payment reform for PSM.

Variable	Dingyuan (*N* = 1552)		Weiyuan (*N* = 1486)	
	Mean	SD	Mean	SD
AA	1.44	0.50	1.45	0.50
AD	1.29	0.45	1.07	0.26
Hospital level	1.57	0.50	1.44	0.50
Gender	1.47	0.50	1.56	0.50
Age	56.20	22.46	45.81	20.96
Occupation	5.01	0.20	5.14	0.75
Marital status	1.85	0.37	1.83	0.45
Admission department	1.58	0.87	1.26	0.76
Admission route	1.30	0.46	1.30	0.46
Length of stay	6.11	4.68	6.19	1.86
Admission status	1.31	0.55	1.58	0.51
Health status	1.34	0.63	1.69	0.51
History of disease	0.46	0.50	0.19	0.39
Chronic	0.40	0.49	0.14	0.34
History of surgery	0.05	0.21	0.01	0.10

AA, the appropriateness of admission; AD, the appropriateness of disease. The mean shows the annual average for each variable. SD stands for standard deviations.

**Table 5 ijerph-16-01410-t005:** Overall balance test and matching results from three kinds of matching method.

Sample	Overall Balance					Matching Results			Bootstrap Results			
	Pseudo R^2^	LR chi^2^	P > chi^2^	Mean Bias	Median Bias	ATT	S.E.	T-stat	S.E.	*z*-Value	*p*-Value	95%CI (lower, upper)
AA												
Raw sample before matching	0.275	1321.76	0.000	32.2	25.7	−0.01	0.02	−0.68				
kernel matching	0.003	16.50	0.927	2.8	2.0	−0.03	0.03	−2.05	**0.03**	**−2.42**	**0.042**	−0.08 to 0.02
k-nearest neighbor matching (k = 1)	0.004	22.07	0.896	3.7	2.8	−0.04	0.03	−2.18	**0.04**	**−2.62**	**0.038**	−0.11 to 0.03
local linear regression matching	0.006	19.16	0.919	3.3	2.6	−0.02	0.03	−1.59	0.04	−1.77	0.073	−0.07 to 0.02
AD												
Raw sample before matching	0.233	879.57	0.000	30.2	21.3	0.22	0.01	16.57				
kernel matching	0.003	3.42	0.943	2.2	1.4	0.19	0.03	13.87	**0.02**	**14.28**	**0.000**	0.17 to 0.25
k-nearest neighbor matching (k = 1)	0.012	4.06	0.933	2.9	2.1	0.17	0.05	11.15	**0.04**	**11.56**	**0.000**	0.15 to 0.19
local linear regression matching	0.019	12.79	0.712	5.4	5.6	0.16	0.02	11.00	**0.02**	**11.48**	**0.000**	0.14 to 0.20

AA, the appropriateness of admission; AD, the appropriateness of disease; S.E., standard error; CI, confidence interval. All results are computed using the Stata module of psmatch2. The S.E. reported by matching result did not consider the fact that the propensity score was estimated (that is, assuming the propensity score is the true value, and then deducing the S.E.), so we considered to get a more accurate S.E. and 95%CI using the bootstrap method.

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
