# Peer review of "Is There a Difference in the Utilisation of Inpatient Services Between Two Typical Payment Methods of Health Insurance? Evidence from the New Rural Cooperative Medical Scheme in China"

_ijerph, 2019, doi:10.3390/ijerph16081410_

Round 1
Reviewer 1 Report
This manuscript is written well. However, this need to revise several points.
1) Figures are not easy to understand because of too small.
This need to revise.
2) English langage is need to revise for whole of manuscript.
Coudld you consider this?
Author Response
Replies to reviewer 1’s comments:
Dear Reviewer,
Thanks for your wonderful comments and suggestions on our manuscript. We have taken into account the comments and suggestions from you, in which we found most helpful. We are pleased to response to them point by point and changes in the article. We hope that the revised manuscript will satisfy you.
Comment 1:
1. Figures are not easy to understand because of too small. This need to revise.
Response: Thank you for your kind reminder. It is true that the size of the previous picture is too small, which is not conducive to editing and publishing and readers' reading. I have turned the picture up.
Comment 2:
2. English langage is need to revise for whole of manuscript. Could you consider this?
Response: Thank you for your kind comments. I have invited two native English speaking professionals to revise the language of this article.
Reviewer 2 Report
This article offers methodologies and parameters to assess the impact of different payment methods on the utilization of impatient services in China, before and after their reform.
The non-expert readers may have difficulties in understanding the existing relationships among policies, system of funding and reimbursement, decision making and behavior at individual, local hospital and county. The authors consider only quantitative analysis and avoid qualitative methods. A better description of the different system of payment at individual, local hospital and county before statistical analysis could help to verify the hypothesis of their effectiveness.
Author Response
Replies to reviewer 2’s comments:
Dear Reviewer,
Thanks for your wonderful comments and suggestions on our manuscript. We have taken into account the comments and suggestions from you, in which we found most helpful. We are pleased to response to them point by point and changes in the article. We hope that the revised manuscript will satisfy you.
Comment:
The non-expert readers may have difficulties in understanding the existing relationships among policies, system of funding and reimbursement, decision making and behavior at individual, local hospital and county. The authors consider only quantitative analysis and avoid qualitative methods. A better description of the different system of payment at individual, local hospital and county before statistical analysis could help to verify the hypothesis of their effectiveness.
Response: Thank you for your kind reminder. First of all, thank you for your very valuable advice. You mentioned that for non-professional readers, there may be some doubts in this professional field. However, this paper actually introduces the differences between the two regional payment systems in various positions. Firstly, in the Introduction section, a brief introduction is made to hospital-based payment and regional or medical alliance based payment. The above two payment systems actually correspond to the characteristics of Weiyuan and Dingyuan respectively. In addition, in the Method part, table 1 introduces the details of the specific payment system design of the two counties. Finally, in the Discussion part, according to the analysis results, the specific payment system differences of the two counties are further indicated in the cause analysis. Therefore, throughout the whole article, the payment system for individuals, local hospitals and counties has been fully introduced, and we hope that the reviewers can understand it.
Reviewer 3 Report
Abstract is too long. More words then it is allowed by the Publisher.
Literature review and references should be more extensive.
line 48 - "Evidence from many countries, such as South Africa, South Korea and Ghana [2-4]South Africa, South Korea and Ghana" - China has a different economic growth so, maybe the Authors should make comparison with some other regions / countries;
Clear definition of rural and urban area should be provided.
More information about chosen counties / samples - place them in the context of whole China.
Results and discussion should be presented in more clear way.
Author Response
Replies to reviewer 3’s comments:
Dear Reviewer,
Thanks for your wonderful comments and suggestions on our manuscript. We have taken into account the comments and suggestions from you, in which we found most helpful. We are pleased to response to them point by point and changes in the article. We hope that the revised manuscript will satisfy you.
Comment 1:
1. Abstract is too long. More words then it is allowed by the Publisher.
Response: Thank you for your kind reminder. I've reduced the abstract to 211 words, which helps with the publication format.
Comment 2:
2. Literature review and references should be more extensive.
Response: Thank you for your kind comments. Thank you very much for your valuable comments. I have supplemented the literature review of Introduction part and added references in other parts.
Comment 3:
3. line 48 - "Evidence from many countries, such as South Africa, South Korea and Ghana [2-4]South Africa, South Korea and Ghana" - China has a different economic growth so, maybe the Authors should make comparison with some other regions / countries;
Response: Thank you for your kind comments. Due to the consumer can choose different levels of medical institutions seek medical service in above three countries and China, the patients have the similiar medical behavior characteristics, which are very different with some developed countries such as Britain. In Britain, the government and medical provider forced consumer to choose primary medical institutions, so this study uses some study results in the above three countries to compared with China, and the macroeconomic characteristics such as different economic growth in some countries maybe cannot affect the behavior of medical supply and demand in both sides, so this sentence have comparability and the scientific nature, we hope reviewer can understand.
Comment 4:
4. Clear definition of rural and urban area should be provided.
Response: Thank you for your kind reminder. The main research scope of this paper is rural areas. For China's medical service system, rural areas mostly refer to service providers and consumers within the county. Therefore, I added the definition of rural areas in the first paragraph of Introduction.
“The medical service system in China's rural areas is usually defined as the region where the three-level rural medical service network is located, namely the county scope, which includes the three-level rural medical institutions in the county, and the service objects should be consumers within the county scope.”
Comment 5:
5. More information about chosen counties / samples - place them in the context of whole China.
Response: Thank you for your kind reminder. We very much agree with your valuable opinion that I have added Table 1 to show the basic economic and health conditions of the chosen counties. Please see Table 4 for sampling information. Thank you!
Comment 6:
6. Results and discussion should be presented in more clear way.
Response: Thank you for your kind suggestion. We agree with you very much, so we have simplified the redundancy of the results and rearrange the logic of the discussion sections.
Round 2
Reviewer 3 Report
I accepted correction. They improved the scientific quality of the article.